# Efficacy and Safety of a Novel Therapeutic of Natural Origin (NTN) in Adult Patients with Lactose Intolerance: A Multicenter, Randomized, Crossover, Double-Blind, Placebo-Controlled Study

**DOI:** 10.3390/foods11172600

**Published:** 2022-08-26

**Authors:** Corina Pop, Ioan Sporea, Javier Santos, Nicolae Tudor, Nicoleta Tiuca

**Affiliations:** 1Department of Gastroenterology and Internal Medicine, Clinical Emergency University Hospital, Bucharest, Romania; 2Department of Gastroenterology, Clinical Emergency University Hospital, Timisoara, Romania; 3Laboratory of Neuro-Immuno-Gastroenterology, Digestive System Research Unit, Vall d’Hebron Institut de Recerca (VHIR), Vall d’Hebron Hospital Universitari, Barcelona, Spain; 4Department of Gastroenterology, Vall d’Hebron Hospital Universitari, Barcelona, Spain; 5Centro de Investigación Biomédica en Red de Enfermedades Hepáticas y Digestivas (CIBERHED), Instituto de Salud Carlos III, Madrid, Spain; 6Department of Gastroenterology, Elias University Clinical Hospital, Carol Davila University, Bucharest, Romania; 7Department of Internal Medicine, Clinical Emergency University Hospital, Str. Splaiul Independentei 169, Sector 5, Bucharest, Romania

**Keywords:** lactose intolerance, novel therapeutic of natural origin, hydrogen breath test, food intolerance, intestinal barrier

## Abstract

Background: Film-forming substances, such as natural polysaccharides (NP) and pea proteins (PP), act as a protective barrier for treating various gastrointestinal conditions. We assessed the efficacy and safety of a novel therapeutic of natural origin (NTN) containing NP and PP for symptomatic treatment of lactose intolerance. Methods: In this multicenter, randomized, double-blind, parallel-group study, patients with lactose intolerance received NTN (*n* = 30) or placebo (*n* = 30) for 7 days, then the alternate treatment for 7 days. Patients rated their gastrointestinal symptoms using a 7-point Likert scale. The lactose hydrogen breath test was used to assess exhaled hydrogen. Results: NTN as primary or crossover treatment significantly improved patient-reported symptoms of bloating, distension, and abdominal pain. Abdominal pain also improved under primary treatment with placebo. Primary treatment with NTN, but not placebo, normalized mean exhaled hydrogen levels. In the group allocated initially to placebo, crossover to NTN attenuated the increase in hydrogen production. No treatment-related adverse effects were reported in either group. Conclusions: Subjective improvements in bloating, distension, and abdominal pain with NTN were supported by objective evidence of hydrogen production normalization. NTN appears to be a useful alternative to lactose avoidance or enzyme replacement in patients with lactose intolerance.

## 1. Introduction

Food intolerance affects up to 20% of the population [1,2], with lactose intolerance being the most common form. In contrast to food allergies, which are immune mediated [3], etiologies of food intolerances include non-coeliac gluten/wheat sensitivity, enzyme/transport defects, and/or pharmacological activity of substances present in food or food additives [1,2]. Despite the different pathogenic mechanisms, food intolerance and food allergy can be difficult to separate due to overlap in certain triggers (e.g., wheat, milk) and in signs/symptoms [1,3]. Diagnosis is often not straightforward and requires an understanding of the varied clinical presentations, including the severity and timing of symptom onset [1]. Identifying food-intolerant patients for inclusion in clinical trials can be extremely challenging.

Lactose malabsorption occurs when lactase (β-galactosidase) activity is reduced in the brush border of the small intestine [4,5], frequently leading to symptoms indicative of lactose intolerance such as bloating, flatulence, and abdominal pain [1,6,7,8]. A genetically programmed decrease in lactase synthesis during adulthood (lactase nonpersistence) is responsible for primary lactase deficiency, which affects about two-thirds of the adult population [9,10]. Secondary deficiency arises from injury to the intestinal mucosa (e.g., Crohn’s disease, coeliac disease, infections), which reduces the amount of available lactase [6,7,9].

Reduced lactase activity increases the presence of lactose in the gut lumen and promotes its fermentation by the gut microbiota to release gases (hydrogen, methane, carbon dioxide) and short-chain fatty acids, resulting in increased colonic distention and accelerated orocecal transit time. Other factors influencing the efficiency of lactose metabolism include gastric emptying and intestinal transit times and/or response of the intestinal tract to an increased osmotic load [11].

A diagnosis of lactose intolerance is often made clinically and in response to an empirical trial of dietary lactose avoidance. The condition can also be diagnosed using a breath test based on the production of hydrogen (plus carbon dioxide and methane) by intestinal microbiota following fermentation of undigested lactose; the gases are absorbed and eliminated by the lungs [1,6]. Another diagnostic method is genetic testing which can be used to identify a polymorphism (LCT-13910C>T) in intron 13 of the *MCM6* gene associated with lactase persistence [6].

The main therapeutic intervention for lactose intolerance is dietary restriction, followed by gradual reintroduction of lactose-containing foods [1]. This approach is complicated, however, by the widespread use of lactose as a food additive or so-called ‘hidden’ lactose [1]. Other approaches that benefit some individuals include taking lactase supplements (prior to a lactose-containing meal) or specific probiotic strains capable of expressing β-galactosidase enzymatic activity [9]. Evidence is scarce for any benefit associated with colonic adaptation (regular administration of increasing quantities of lactose) or administration of rifaximin, a nonabsorbable antibiotic [9]. A need thus exists for innovative options that can control symptoms effectively without negative side effects, improve patients’ quality of life, and are suitable for use in a broad range of patients. 

A novel therapeutic of natural origin (NTN) is indicated for symptomatic management of food maldigestion or food intolerance. NTN contains a mixture of naturally occurring polysaccharides, pea proteins (*Pisum sativum*), tannins derived from grape seed extract, and β-galactosidase. Pea protein is a mucomimetic substance that creates a synergistic mechanical barrier which has an emollient and soothing action on the digestive tract [12,13]. The proanthocyandins found in grape seed extract are highly antioxidant [14]. The beneficial effects of these substances have previously been demonstrated in a murine model of vulvovaginal candidiasis [15] and in clinical studies of diarrhea-predominant irritable bowel syndrome [16,17]. β-galactosidase has wide application in the food industry to manufacture lactose-hydrolyzed products for persons with lactose intolerance [9,18]. In a murine model of fructose, carbohydrate, and fat intolerance, NTN was shown to significantly improve gut homeostasis and organ function [19]. The NTN formulation also contains strains of tyndallized *Lactobacillus reuteri*, and tyndallized *L. acidophilus*, which prevent and reduce symptoms associated with altered gut flora and lactose intolerance, respectively [20]. NTN aims to control symptoms in food-intolerant individuals by restoring intestinal mucosal integrity and normalizing carbohydrate and cholesterol metabolism together with antioxidative activity. 

This placebo-controlled study was conducted to investigate the efficacy and safety of NTN for the treatment of lactose intolerance. 

## 2. Materials and Methods

### 2.1. Study Design

This phase IV, multicenter, randomized, parallel-group, double-blind study was performed at gastroenterology outpatient medical centers in Romania. The crossover design involved 7 days’ treatment with NTN or placebo followed by 7 days’ treatment with the alternate regimen. Six study visits occurred: Visit 1—Day 0, Visit 2—Day 1, Visit 3—Day 3, Visit 4—Day 7, Visit 5—Day 8, and Visit 6—Day 14 (Figure 1).

### 2.2. Selection Criteria

Eligible for inclusion were adults of Caucasian ethnicity with a current or recently reported history of dairy intolerance of at least 1 month’s duration who were willing to sign the informed consent form. Subjects were recruited by gastroenterologists or general medicine physicians at individual clinics. 

The main exclusion criteria were: diagnosis of coeliac disease, gastroenteritis, Crohn’s disease, or diabetes; use of lactase enzyme tablets or drops; use of antibiotics in the previous 4 weeks; use of laxatives in the previous 2 weeks, including for colonoscopy; pregnant or breastfeeding women; and allergy to any of the product ingredients.

### 2.3. Patients and Treatment 

Using a computer-generated sequence, patients were randomized 1:1 to receive NTN (Novinthetical, Lugano, Switzerland) or placebo for 7 days, then crossed over to receive the alternate treatment for 7 days. Patients were instructed to take one capsule of NTN before each main meal per day (i.e., three capsules per day). Patients were advised to not consume more than 500 mL of dairy products daily. 

### 2.4. Efficacy Endpoints 

Primary efficacy endpoints were the evolution of gastrointestinal symptoms in response to study treatment, and the change in total hydrogen production in response to lactose challenge, during primary and crossover treatment. Gastrointestinal symptoms of bloating (sensation of abdominal swelling), distension (increase in measured abdominal size), and abdominal pain were self-evaluated by patients using a 7-point Likert scale scored from ‘very bad’ to ‘very good’. 

To confirm the presence of lactose malabsorption, patients underwent a hydrogen breath test (HBT). Patients were given a drink containing 50 g of lactose. Exhaled hydrogen was measured before lactose ingestion, then every 30 min up to 120 min after lactose ingestion (normal value <20 ppm). Comparisons were made for changes in 2 h total hydrogen production between Days 1 and 7, Days 8 and 14, and Days 1 and 14 of study treatment. 

The secondary endpoint of the study was NTN safety, which was assessed by monitoring the occurrence of undesirable effects reported by the subject or observed by the physician during the study period.

Data were collected at baseline (Day 1 of primary treatment), on Days 2 and 7 of primary treatment, and on Days 8 and 14 of crossover treatment.

### 2.5. Statistical Analysis

Data from all patients who received at least one dose of treatment were analyzed in the intention-to-treat (ITT) analysis. For the efficacy evaluation, all randomized patients who had at least one post-treatment measurement were included in the modified intention-to-treat population (MITT). Patients who completed both the baseline visit and end-of-treatment visit and had no major protocol violations were included in the per-protocol (PP) population. All patients who received at least one dose of the studied medical device were included in the safety analysis.

Descriptive statistics were used. Continuous variables were summarized using mean and standard deviation. Categorical variables were summarized using absolute and relative frequency counts.

Exploratory statistical tests were performed to test for differences between treatment groups in the evolution of clinical symptoms (proportion of patients with good, unchanged or bad symptom scores at study timepoints) and signs of lactose intolerance (mean values for hydrogen production in response to lactose challenge at study timepoints). The Student’s *t*-test was used. A *p*-value of 0.05 was considered to be statistically significant. 

## 3. Results

Sixty adult patients were enrolled at four gastroenterology clinics in Romania (Bucharest, Timisoara) and randomized to primary treatment with NTN (*n* = 30) or placebo (*n* = 30) before crossover. The demographic and baseline characteristics of the study population are shown in Table 1. The mean age was approximately 43 years (range: 29–58 years). Apart from a higher proportion of females in the NTN/placebo group versus the placebo/NTN group (77% vs. 57%), there were no notable differences between the two groups. No patient in either group had any known allergies. One patient randomized to primary treatment with placebo discontinued the study after one dose of study treatment. All remaining patients completed the study as planned.

### 3.1. Efficacy

The evolution of bloating in patients randomized to NTN (Days 1 to 7) with crossover to placebo (Days 8 to 14) and in patients randomized to placebo with crossover to NTN is shown in Figure 2. In the group treated initially with NTN (Figure 2a), the proportion of patients with bad Likert scores decreased, and the proportion of patients with good Likert scores increased from Day 1 to Day 7. On Day 2, an additional 23% of patients reported ‘very good, better or somewhat good’ bloating symptoms compared with Day 1 (*p* = 0.01130). On Day 7, there was a 67% increase in the proportion of patients reporting ‘very good, better or somewhat good’ bloating symptoms (*p* = 0.00001) and a 96% decrease in the proportion of patients reporting ‘very bad, worse, somewhat bad’ bloating symptoms (*p* = 0.00001) compared with Day 1. After crossover to placebo on Day 8, the proportion of patients reporting good Likert scores decreased, and the proportion of patients reporting bad Likert scores increased on Day 14. In the group allocated to initial treatment with placebo (Figure 2b), the difference between Day 1 and Day 7 in the proportion of patients reporting ‘very good, better or somewhat good’ bloating symptoms was not statistically significant. After crossover to NTN on Day 8, the proportion of patients with good Likert scores increased, and the proportion of patients with bad Likert scores increased. On Day 14, 80% of patients reported ‘very good, better or somewhat good’ bloating symptoms (*p* = 0.00004 vs. Day 8), and 10% of patients reported ‘very bad, worse, somewhat bad’ bloating symptoms. The proportion of patients reporting bad bloating symptoms on Day 14 was 75% lower compared with the start of crossover (*p* = 0.007 vs. Day 8).

The evolution of distension in patients in the NTN/placebo and placebo/NTN groups is shown in Figure 3. In the group treated initially with NTN (Figure 3a), 93.3% of patients reported ‘very bad, worse, somewhat bad’ distension symptoms on Day 1. On Day 2, the proportion of patients reporting ‘very good, better, or somewhat good’ distention symptoms increased by 23% (*p* = 0.01), and the proportion of patients reporting ‘very bad, worse, or somewhat bad’ distention symptoms decreased by 73% (*p* = 0.00001) compared with Day 1. On Day 7 of NTN treatment, 63.3% of patients reported good distension symptoms, and 3.3% of patients reported bad distension symptoms (*p* = 0.00001 vs. Day 1). Following crossover to placebo on Day 8, the proportion of patients with bad Likert scores increased, and the proportion of patients with good Likert scores decreased to Day 14. In the group treated initially with placebo (Figure 3b), crossover to NTN on Day 8 increased the proportion of patients with good Likert scores and decreased the proportion of patients with bad Likert scores. On Day 14, 76.7% of patients reported ‘very good, better, or somewhat good’ distention symptoms compared with 36.7% at the start of crossover (*p* = 0.002 vs. Day 8), and no patients reported bad symptoms compared with 30% at the start of crossover (*p* = 0.006 vs. Day 8).

Figure 4 shows the evolution of abdominal pain in groups receiving NTN/placebo or placebo/NTN. When administered as primary treatment, both NTN (Figure 4a) and placebo (Figure 4b) increased the proportion of patients with ‘very good, better or somewhat good’ scores for abdominal pain, although the difference between Day 1 and Day 7 was statistically significant for NTN (*p* = 0.044) but not placebo (*p* = 0.054). The difference between Day 1 and Day 14 in the proportion of patients with ‘very good, better, or somewhat good’ abdominal pain scores was significant for both the NTN/placebo group (*p* = 0.035) and the placebo/NTN group (*p* = 0.048).

The change in exhaled hydrogen before and after lactose ingestion is shown in Table 2 and Table 3. On Day 1, patients allocated to NTN/placebo (Table 2) were positively diagnosed with lactose intolerance based on a mean change (Δ) in peak basal HBT of 21.3 ppm after lactose ingestion. During primary treatment with NTN, 2 h hydrogen production after lactose ingestion was lower, and patients tested negative for lactose intolerance (i.e., the mean increase in HBT was <20 ppm). Mean hydrogen production increases after lactose ingestion was 16.7 ppm on Day 2 and 8.8 ppm on Day 7 (*p* = 0.02 vs. Day 1). After crossover to placebo, patients continued to show hydrogen production within the normal range after lactose ingestion, with mean values of 12.7 ppm on Day 8 and 14.2 ppm on Day 14. The group allocated to placebo/NTN (Table 3) was positively diagnosed with lactose intolerance on Day 1 based on a Δ peak basal HBT of 20.2 ppm. During primary treatment with placebo, 2 h hydrogen production after lactose ingestion was higher, and patients continued to test positive for lactose intolerance (i.e., the mean increase in HBT was >20 ppm). Mean hydrogen increases after lactose ingestion were 45.7 ppm on Day 2 (*p* = 0.0065 vs. Day 1) and 48.4 ppm on Day 7 (*p* = 0.009 vs. Day 1). After crossover to NTN, the increase in hydrogen production after lactose ingestion was significantly attenuated, with mean values of 20.9 ppm on Day 8 and 5.5 ppm on Day 14 (*p* = 0.0009 vs. Day 8). 

### 3.2. Safety

During the study, one adverse event was reported for a single patient in the placebo/NTN arm. After the first dose of placebo, the patient presented intense abdominal pain and diarrhea, which led to treatment discontinuation. The event was considered to be related to the underlying condition of lactose intolerance, not to the study medication. 

## 4. Discussion

Food intolerance is often generated by a partial or total loss of a person’s ability to digest certain food substances due to enzyme or transport defects, resulting in colonic fermentation and increased gas production, which give rise to gastrointestinal symptoms [1]. There is also evidence in the literature linking food intolerances to intestinal barrier dysfunction [21]. These findings provide a rationale for considering mucosal protectors as potential therapies. 

Our study demonstrated that NTN as a primary treatment (comparison between Day 1 and Day 7) or as a crossover treatment (comparison between Day 1 and Day 14), produced statistically significant improvement in bloating and distension in patients with lactose intolerance. NTN, but not placebo, as primary treatment significantly improved abdominal pain from Day 1 to Day 7. In the Day 1 to Day 14 comparison, significant improvements in abdominal pain were observed irrespective of whether patients had started treatment with NTN and crossed over to placebo or had started treatment with placebo and crossed over to NTN. The reasons for the apparent response of abdominal pain to placebo treatment warrant further investigation.

Patient-reported outcomes have been developed to better capture patients’ subjective experience of an intervention. In our study, gastrointestinal symptoms were self-assessed by patients using a 7-point Likert scale. This patient-reported Global Improvement Scale has been validated in irritable bowel syndrome [22], a condition with symptoms similar to those of lactose and other food intolerances. The efficacy of NTN in improving patients’ perception of their gastrointestinal symptoms was supported by the lactose HBT results, an objective test based on the fermentation of undigested lactose by bacterial microbiota [23]. An increase in exhaled hydrogen of ≥20 ppm above the baseline value in two consecutive readings indicates small intestinal bacterial overgrowth (SIBO), i.e., the presence of abnormally high numbers of colonic bacteria in the small intestine [23]. Patients’ inability to hydrolyze and absorb lactose was confirmed at baseline (Day 1) using the HBT, with both groups showing mean increases in hydrogen production of ≥20 ppm after lactose ingestion. Irrespective of whether NTN was administered as primary or crossover treatment, exhaled hydrogen production after lactose ingestion was lowered to within the normal range. Notably, when NTN was administered as primary treatment, attenuation of hydrogen production after lactose ingestion was maintained even after patients had crossed over to receive placebo for 7 days.

The absence of any adverse events, serious adverse events, or serious unexpected severe adverse reactions with NTN in this study supports its excellent safety profile. The single adverse event reported in a patient receiving a placebo was considered unrelated to treatment. 

A major strength of the study is its well-controlled multicenter design, which is more likely to improve provider performance and impact positively on patient outcomes [24]. The crossover method allows for the inclusion of fewer patients and limits the influence of potential confounding variables since patients act as their own controls. Evaluation of interventions within the same patient eliminates between-subject variability [25]. Study limitations include the modest sample size and the possibility of carryover effects due to the lack of washout between treatment phases. As patients were included based on a ‘history of dairy intolerance’, it is uncertain whether all participants were truly lactose intolerant or presented a mix of digestive disorders (e.g., lactose maldigestion, irritable bowel syndrome). The results also may have been influenced by allowing for continued dairy ingestion (up to 500 mL/day) throughout the study, as this can induce colonic bacterial adaptation. The relatively low peaks of hydrogen production after the first HBT on Day 1, which were at the threshold for a positive diagnosis of lactose intolerance, suggest the possibility of colonic adaptation and/or inclusion of patients without true lactose intolerance. 

Notwithstanding the limitations, NTN was shown to provide significant relief of gastrointestinal symptoms within 1 day of treatment. Patients’ subjective perception of clinical improvement was supported by objective evidence from the HBT. The HBT has become a popular technology to aid in the diagnosis of lactose malabsorption/intolerance on account of its relatively low cost, availability, non-invasiveness, and diagnostic efficiency, with a mean sensitivity of 77.5% and a specificity of 97.6% [26,27]. The test may also be used to understand abnormal pathophysiology such as SIBO and carbohydrate malabsorption contributing to symptoms in patients presenting with food intolerances. A possible limitation of the HBT is false negative results in individuals with fixed hydrogen and non-methane production, although these constitute a low proportion (3.4%) of breath tests conducted [28].

Despite the high prevalence of food intolerances in the general population, treatment options are limited. NTN is one among a suite of novel reticulated pea protein-containing mucoprotectors with indications for use in the gastrointestinal tract, urinary tract, vaginal tract, and respiratory tract [29]. Our study demonstrated that NTN administered to patients with lactose intolerance produced significant improvement in patient-reported bloating, distension, and abdominal pain and normalized exhaled hydrogen in the lactose HBT. NTN was well tolerated. The results support its use for symptomatic improvement in patients with lactose intolerance. The effectiveness of this novel therapy for lactose intolerance allows us to hypothesize that NTN may provide similar benefits for other food intolerances with a similar etiology.

## Figures and Tables

**Figure 1 foods-11-02600-f001:**
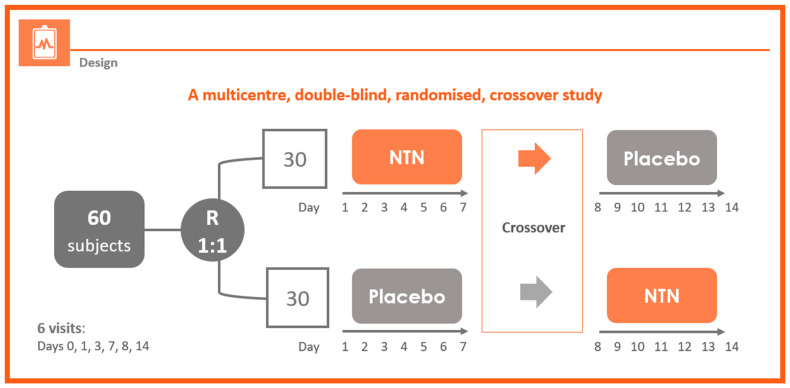
Study design.

**Figure 2 foods-11-02600-f002:**
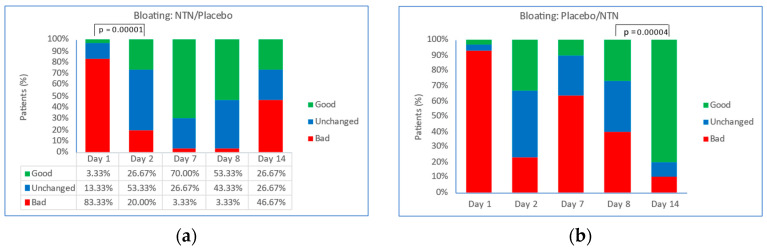
Evolution in bloating. Proportion of patients with good, unchanged, or bad Likert scores in groups randomized to receive: (**a**) NTN on Days 1–7 with crossover to placebo on Days 8–14; *p* = 0.00001, significant decrease in bad bloating feeling at Day 7; (**b**) placebo on Days 1–7 with crossover to NTN on Days 8–14; *p* = 0.00004, significant increase in good bloating feeling at Day 14. Good: somewhat good/good/very good; Unchanged: same; Bad: somewhat bad/bad/very bad; NTN: novel therapeutic of natural origin.

**Figure 3 foods-11-02600-f003:**
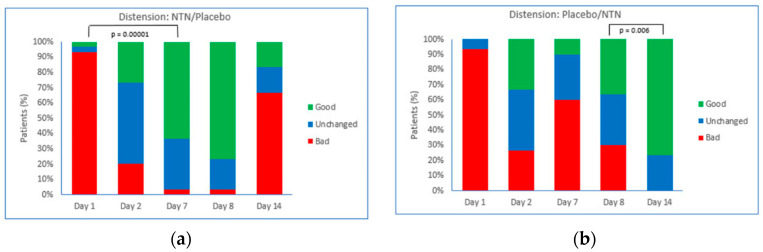
Evolution of distension. Proportion of patients with good, unchanged, or bad Likert scores in groups randomized to receive: (**a**) NTN on Days 1–7 with crossover to placebo on Days 8–14; *p* = 0.00001, significant decrease in bad distension feeling at Day 7; (**b**) placebo on Days 1–7 with crossover to NTN on Days 8–14; *p* = 0.006, significant decrease in bad distension feeling at Day 14. Good: somewhat good/good/very good; Unchanged: same; Bad: somewhat bad/bad/very bad; NTN: novel therapeutic of natural origin.

**Figure 4 foods-11-02600-f004:**
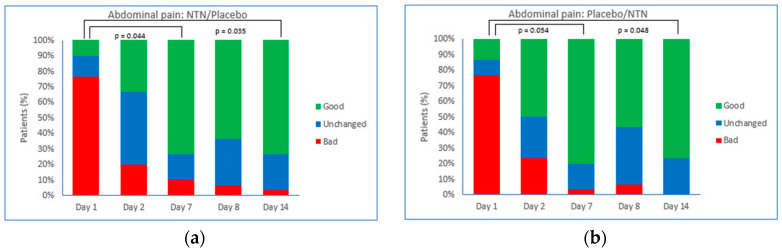
Evolution of abdominal pain. Proportion of patients with good, unchanged, or bad Likert scores in groups randomized to receive: (**a**) NTN on Days 1–7 with crossover to placebo on Days 8–14; *p* = 0.044, significant increase in good abdominal pain feeling at Day 7. At study end, significantly more patients reported good symptoms (*p* = 0.035); (**b**) placebo on Days 1–7 with crossover to NTN on Days 8–14; *p* = 0.054, no significant increase in good abdominal pain feeling at Day 7. At study end, significantly more patients reported good symptoms (*p* = 0.048). Good: somewhat good/good/very good; Unchanged: same; Bad: somewhat bad/bad/very bad; NTN: novel therapeutic of natural origin.

**Table 1 foods-11-02600-t001:** Baseline demographic and clinical characteristics.

Characteristic	NTN (*n* = 30)	Placebo (*n* = 30)
Sex, M/F	7 (23) / 23 (77)	13 (43)/17 (57)
Age, years	43.6 ± 11.7	43.4 ± 14.5
Weight, kg	79.4 ± 10.7	73.1 ± 20.9
Height, cm	161.1 ± 8.8	161.9 ± 33.4
BMI, kg/m^2^	28.4 ± 3.9	25.9 ± 6.9
Comorbidity	5 (16.7)	7 (23.3)

Continuous variables are expressed as mean ± SD, and categorical variables are expressed as numbers (%). BMI: body mass index; F: female; M: male; NTN: novel therapeutic of natural origin; SD: standard deviation.

**Table 2 foods-11-02600-t002:** Evolution of the hydrogen breath test in patients receiving NTN on Days 1–7 with crossover to placebo on Days 8–14.

Parameter	NTN			Placebo	
	Day 1	Day 2	Day 7	Day 8	Day 14
HBT basal	22.3 ± 4.51	23.0 ± 4.08	22.5 ± 4.24	21.4 ± 3.71	19.5 ± 4.24
HBT peak	43.5 ± 3.74	39.7 ± 3.45	31.3 ± 4.50	34.1 ± 4.09	33.7 ± 4.87
Δ peak basal HBT	21.3	16.7	8.8 *	12.7	14.2

Results are expressed as mean hydrogen (H_2_) level (ppm). * Statistically significant difference versus Day 1, *p* = 0.02. HBT: hydrogen breath test; NTN: novel therapeutic of natural origin.

**Table 3 foods-11-02600-t003:** Evolution of the hydrogen breath test in patients receiving placebo on Days 1–7 with crossover to NTN on Days 8–14.

Parameter	Placebo			NTN	
	Day 1	Day 2	Day 7	Day 8	Day 14
HBT basal	22.9 ± 4.56	23.5 ± 4.09	22.8 ± 4.25	21.6 ± 3.69	19.4 ± 4.23
HBT peak	43.1 ± 3.81	69.2 ± 3.27	71.2 ± 4.46	42.5 ± 3.91	24.9 ± 4.84
Δ peak basal HBT	20.2	45.7	48.4	20.9	5.4 *

Results are expressed as mean hydrogen (H_2_) level (ppm). * Statistically significant difference versus Day 8, *p* = 0.0009. HBT: hydrogen breath test; NTN: novel therapeutic of natural origin.

## Data Availability

Data are available on reasonable request.

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
