# Peer review of "Efficacy and Safety of a Novel Therapeutic of Natural Origin (NTN) in Adult Patients with Lactose Intolerance: A Multicenter, Randomized, Crossover, Double-Blind, Placebo-Controlled Study"

_foods, 2022, doi:10.3390/foods11172600_

Round 1

Reviewer 1 Report

I have reviewed the article by Pop C et al. “Efficacy and safety of a novel therapeutic of natural origin (NTN) in adult patients with lactose intolerance: a multicenter, randomized, crossover, double-blind, placebo-controlled study”

 The authors use a Swiss product made up of natural polysaccharides and pea protein which have a mucous barrier protective effect and may reduce symptoms related to barrier defects. The authors have previously published the use of this or similar product in patients with IBS-D. Now a similar study is carried out in persons with lactose intolerance. The results seem to support the view that in this double blind controlled cross over study the outcomes measured by subjective symptomatic reductions and diminished breath hydrogen response to 50 g lactose challenge.

General Comments:

The concept is interesting and the results support the authors hypotheses that symptoms improve.

 In my opinion there are two criticisms of the study. The first is the selection of patients with a history of dairy intolerance. This group therefore constitutes self -reported lactose intolerance. As such it is a heterogeneous group of people withl lactose maldigestion, lactose digestion capability, irritable bowel syndrome as well as possible intolerance to genetic A1milk. Therefore the diagnosis of lactose intolerance in the study group is somewhat uncertain.

 In the same vein and second criticism, 500ml of dairy ingestion was allowed throughout the study, but we do not really know the amount /day that was ingested by each person. As such continued dairy ingestion may have induced colonic bacterial adaptation which might have biased results (for example the similar response between NTN and placebo between days 1 and 14 described on pg 5 second paragraph starting with the sentence `Figure 4`. In addition the relatively low peaks of hydrogen in both Table 2 and Table 3 on day 1 could support some degree of adaptation or inclusion of participants without true lactose maldigestion.

 I think these considerations should be included in discussion of limitations

 Minor comments

 Pg 2 in the introduction 2nd paragraph listing ulcerative colitis with cause of secondary lactase deficiency is not quite accurate since colitis does not affect the digestion of lactose. In ulcerative colitis lactose maldigestion is rather dependent on ethnicity.

 Pg 3 section 2.4 It may be useful to just emphasize in the first sentence that the symptoms and hydrogen production were in response to lactose challenge

 Pg 4 Under Efficacy: The authors could define the difference between the sensations of bloat and distension (was this a measured variable?)

 Pg 13 Table 3 box at the bottom with Day 1 please correct the number supplied

Author Response

Comments and Suggestions for Authors

I have reviewed the article by Pop C et al. “Efficacy and safety of a novel therapeutic of natural origin (NTN) in adult patients with lactose intolerance: a multicenter, randomized, crossover, double-blind, placebo-controlled study”

 The authors use a Swiss product made up of natural polysaccharides and pea protein which have a mucous barrier protective effect and may reduce symptoms related to barrier defects. The authors have previously published the use of this or similar product in patients with IBS-D. Now a similar study is carried out in persons with lactose intolerance. The results seem to support the view that in this double blind controlled cross over study the outcomes measured by subjective symptomatic reductions and diminished breath hydrogen response to 50 g lactose challenge.

General Comments:

The concept is interesting and the results support the authors hypotheses that symptoms improve.

 In my opinion there are two criticisms of the study. The first is the selection of patients with a history of dairy intolerance. This group therefore constitutes self -reported lactose intolerance. As such it is a heterogeneous group of people withl lactose maldigestion, lactose digestion capability, irritable bowel syndrome as well as possible intolerance to genetic A1milk. Therefore the diagnosis of lactose intolerance in the study group is somewhat uncertain.

 In the same vein and second criticism, 500ml of dairy ingestion was allowed throughout the study, but we do not really know the amount /day that was ingested by each person. As such continued dairy ingestion may have induced colonic bacterial adaptation which might have biased results (for example the similar response between NTN and placebo between days 1 and 14 described on pg 5 second paragraph starting with the sentence `Figure 4`. In addition the relatively low peaks of hydrogen in both Table 2 and Table 3 on day 1 could support some degree of adaptation or inclusion of participants without true lactose maldigestion.

 I think these considerations should be included in discussion of limitations

The Discussion (limitations) has been revised in line with your suggestion.

“As patients were included based on a ‘history of dairy intolerance’, it is uncertain whether all participants were truly lactose intolerant or presented a mix of digestive disorders (e.g. lactose maldigestion, irritable bowel syndrome). The results may have been influenced also by allowing for continued dairy ingestion (up to 500 ml/day) throughout the study, as this can induce colonic bacterial adaptation. The relatively low peaks of hydrogen production after the first HBT on Day 1, which were at the threshold for a positive diagnosis of lactose intolerance, suggest the possibility of colonic adaptation and/or inclusion of patients without true lactose intolerance.”

 Minor comments

 Pg 2 in the introduction 2nd paragraph listing ulcerative colitis with cause of secondary lactase deficiency is not quite accurate since colitis does not affect the digestion of lactose. In ulcerative colitis lactose maldigestion is rather dependent on ethnicity.

Thank you. Ulcerative colitis has been deleted from the list of conditions causing secondary lactase deficiency.

 Pg 3 section 2.4 It may be useful to just emphasize in the first sentence that the symptoms and hydrogen production were in response to lactose challenge

Text revised. The evolution in symptoms was in response to study treatment. Hydrogen production was in response to a lactose challenge.

 Pg 4 Under Efficacy: The authors could define the difference between the sensations of bloat and distension (was this a measured variable?)

Revised in Methods. “Gastrointestinal symptoms of bloating (sensation of abdominal swelling), distension (increase in measured abdominal size) and abdominal pain ...”.

 Pg 13 Table 3 box at the bottom with Day 1 please correct the number supplied

Corrected. Thank you.

Reviewer 2 Report

Comments to the Author

MANUSCRIPT DETAILS

Ms. Ref. No.: foods-1785789

Title: Efficacy and safety of a novel therapeutic of natural origin (NTN) in adult patients with lactose intolerance: a multicenter, randomized, crossover, double-blind, placebo-controlled study

Article Type: Article

JOURNAL: Foods

GENERAL COMMENTS

The objective of this study is to assess the efficacy and safety of a novel therapeutic of natural origin (NTN) containing NP and PP for symptomatic treatment of lactose intolerance.

The interest in this manuscript idea is significant, but the MS lacks many research basics:

- As referred to a “novel therapeutic of natural origin (NTN)” in the Abstract section, this novel NTN formulation should be provided.

- There are not any references in the Material section.

- Although authors referred to conducted “Statistical analyses”, The SDs are absent in both figures and tables.

- The is conflict in results, e.g. Authors referred in Results section that, age (years) between 18 and 65, while in Table (1) Age (years) for NTN group 43.6 ± 11.7 (means 31.9, 55.3), and Placebo group 43.4 ± 14.5 (means 28.9, 57.9) which lacks accuracy.

- The study design has scientific errors that affect the accuracy of concluded outputs  

I advise authors to consider the raised points for accurate outputs for their idea.

Author Response

The objective of this study is to assess the efficacy and safety of a novel therapeutic of natural origin (NTN) containing NP and PP for symptomatic treatment of lactose intolerance.

The interest in this manuscript idea is significant, but the MS lacks many research basics:

- As referred to a “novel therapeutic of natural origin (NTN)” in the Abstract section, this novel NTN formulation should be provided.

We would certainly prefer to provide the full NTN formulation in the Abstract (i.e. also mentioning tannins derived from grape seed extract and β-galactosidase); however, we are limited by the maximum word count of 200 words.

- There are not any references in the Material section.

On further review of Methods, we don’t perceive any value in providing references for such common instruments as a Likert-type scale and hydrogen breath test.

- Although authors referred to conducted “Statistical analyses”, The SDs are absent in both figures and tables.

As the figures represent a categorical variable (percentage of patients with good, unchanged or bad Likert scores at study timepoints), standard deviations were not calculated. The figure legends have been amended for clarity.

- The is conflict in results, e.g. Authors referred in Results section that, age (years) between 18 and 65, while in Table (1) Age (years) for NTN group 43.6 ± 11.7 (means 31.9, 55.3), and Placebo group 43.4 ± 14.5 (means 28.9, 57.9) which lacks accuracy.

The text has been revised accordingly.

- The study design has scientific errors that affect the accuracy of concluded outputs  

I advise authors to consider the raised points for accurate outputs for their idea.